# Structure of a RecT/Redβ family recombinase in complex with a duplex intermediate of DNA annealing

Brian J. Caldwell[1,2], Andrew S. Norris[3], Caroline F. Karbowski [2], Alyssa M. Wiegand[2], Vicki H. Wysocki [1,3] & Charles E. Bell [1,2,3] ✉

Some bacteriophage encode a recombinase that catalyzes single-stranded DNA annealing (SSA). These proteins are apparently related to RAD52, the primary human SSA protein. The best studied protein, Redβ from bacteriophage λ, binds weakly to ssDNA, not at all to dsDNA, but tightly to a duplex intermediate of annealing formed when two complementary DNA strands are added to the protein sequentially. We used single particle cryo-electron microscopy (cryo-EM) to determine a 3.4 Å structure of a Redβ homolog from a prophage of *Listeria innocua* in complex with two complementary 83mer oligonucleotides. The structure reveals a helical protein filament bound to a DNA duplex that is highly extended and unwound. Native mass spectrometry confirms that the complex seen by cryo-EM is the predominant species in solution. The protein shares a common core fold with RAD52 and a similar mode of ssDNA-binding. These data provide insights into the mechanism of protein-catalyzed SSA.

Bacteriophage with dsDNA genomes often encode a recombination system that consists of two proteins: a 5′-3′ exonuclease for resecting DNA ends, and a recombinase for binding to the resulting 3′-overhang and annealing it to a complementary strand from a homologous duplex[1]. The two proteins form a complex that is thought to load the annealing protein directly onto the 3′-overhang as it is formed by the exonuclease[2,3]. The benefit of these recombination systems for the phage has not been firmly established, but possible roles in replication[4], genome packaging[1], promoting genetic diversity[5,6], and CRISPR-evasion[7] have been proposed. These systems are also found in bacterial genomes within cryptic or active prophage[8], and in mobile genetic elements such as integrating conjugative elements[9] and conjugative plasmids[7] where they can contribute to antibiotic resistance and genetic diversity[10]. While their precise roles in biology are still being studied, the proteins of these recombination systems have been widely exploited in powerful methods for bacterial genome engineering known as recombineering and MAGE (multiplex automated genome engineering)[11–13].

The best studied of these recombination systems is the Red system from bacteriophage λ, for which the exonuclease and annealing proteins are λ exo and Redβ, respectively[14]. λ exo ($M_r$ 24.9 kDa) forms a ring-shaped homotrimer that binds to dsDNA ends and processively digests the 5′-strand to form a long 3′-overhang[15,16]. Redβ is a 30 kDa monomer that binds to ssDNA and promotes the annealing of complementary strands[17,18]. It binds weakly to ssDNA, not at all to preformed dsDNA, but tightly to a duplex intermediate of annealing formed when two complementary strands of DNA are added to the protein sequentially[19]. Coupled to this, Redβ exhibits a dynamic oligomerization in forming rings (or split lock washers) on ssDNA, but helical filaments on annealed duplex[20,21].

Redβ belongs to a large group of proteins annotated as the RecT family based on the protein from the *rac* prophage of *E. coli*[22]. The current Pfam database lists 1549 such sequences, predominantly from bacteriophage or prophage genomes, with zero structures[23]. While this family of proteins was originally thought to be distinct from RAD52[24], the primary SSA protein in human cells[25], more recent sequence

[1]Ohio State Biochemistry Program, The Ohio State University, Columbus, OH 43210, USA. [2]Department of Biological Chemistry and Pharmacology, The Ohio State University, Columbus, OH 43210, USA. [3]Department of Chemistry and Biochemistry and Resource for Native MS-Guided Structural Biology, The Ohio State University, Columbus, OH 43210, USA. ✉e-mail: bell.489@osu.edu

comparisons suggest that they are in fact related[21,26,27]. The structure of an 11-mer ring form of the DNA-binding domain of RAD52 has been determined without DNA[28,29] and with a dT40 oligonucleotide to form a substrate complex[30]. However, there is no structure of RAD52 with two complementary strands of ssDNA bound simultaneously, and its overall mechanism of annealing is still unknown.

Here, we have used single-particle cryo-EM to determine a 3.4 Å structure of a homolog of λ-Redβ from the A118 prophage of *Listeria innocua* that we will refer to as LiRecT. The structure reveals a left-handed helical filament of the protein bound to an 83-mer duplex intermediate of DNA annealing. The filaments are similar to those seen previously for λ-Redβ at low resolution by electron microscopy and atomic force microscopy[20,21], but our structure now reveals the fold of the protein, the location of the DNA binding groove, the conformation of the DNA, and the details of the protein-DNA and inter-subunit interactions. The structure confirms the similarity to RAD52, and reveals a common core fold and shared mode of ssDNA-binding.

## Results

### Architecture of the LiRecT-DNA Complex

The LiRecT protein was purified and found to bind to ssDNA and form a complex with annealed duplex in a similar manner as λ-Redβ, both in phosphate buffered saline (pH 7.4) and in a buffer that was previously used for negative stain EM of λ-Redβ (10 mM $KH_2PO_4$, 10 mM $MgCl_2$, pH 6.0)[20] (Supplementary Fig. 1). For cryo-EM analysis in the latter buffer, a complex of LiRecT with duplex intermediate was formed by incubating the protein with two complementary 83-mer oligonucleotides that were added to the protein sequentially. The sequences of the oligonucleotides were derived from a naturally occurring sequence in M13 DNA described previously[19,31]. The complex appeared as helical filaments of varying lengths, including some with end-on views (Fig. 1a). Standard single-particle analysis without helical symmetry averaging in cryoSPARC yielded a 3.4 Å reconstruction with fully interpretable density for the LiRecT subunits and the bound DNA at the central portion of the filament. The single particle workflow is shown in Supplementary Figs. 2 and 3 and the data collection and refinement statistics are shown in Supplementary Table 1. A captioned PyMOL movie of the structure is provided in Supplementary Movie 1.

In the complex, LiRecT assembles into a left-handed helical filament that is highly reminiscent of those seen previously for λ-Redβ[20]. The filament has an open corkscrew-like shape with an inner diameter of 20 Å, an outer diameter of 100 Å, and a pitch of 105 Å with approximately 10 subunits per turn. The two complementary 83-mer strands are bound as a highly extended and un-wound duplex to a deep, narrow, positively-charged groove that runs along the outer surface of the filament (Fig. 2). One strand, which we call "inner" and

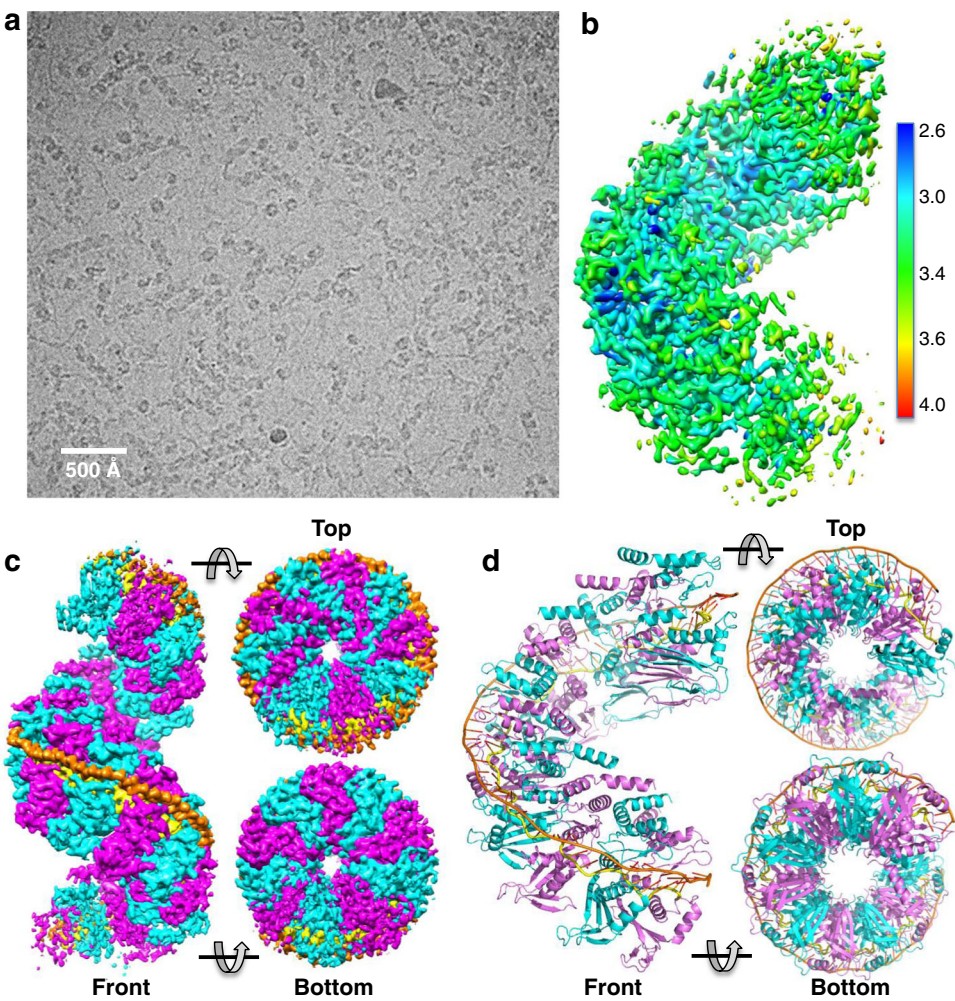

**Fig. 1 | Cryo-EM structure of LiRecT bound to 83-mer annealed duplex.**
**a** Example cryo-EM image at 81,000×. The image is one of 2038 that gave similar results. **b** 3D reconstruction colored according to local resolution. The map covers the central 10 subunits of the filament. **c** Views of the full 18-subunit 3D reconstruction with alternating LiRecT subunits colored cyan and magenta. The inner DNA strand is yellow and the outer strand is orange. Notice that the density gets progressively weaker towards the filament ends. **d** Ribbon diagrams of the 10-subunit model fit and refined to the density for the central portion of the filament. Notice that the duplex is bound in an unusual conformation that is highly extended and un-wound.

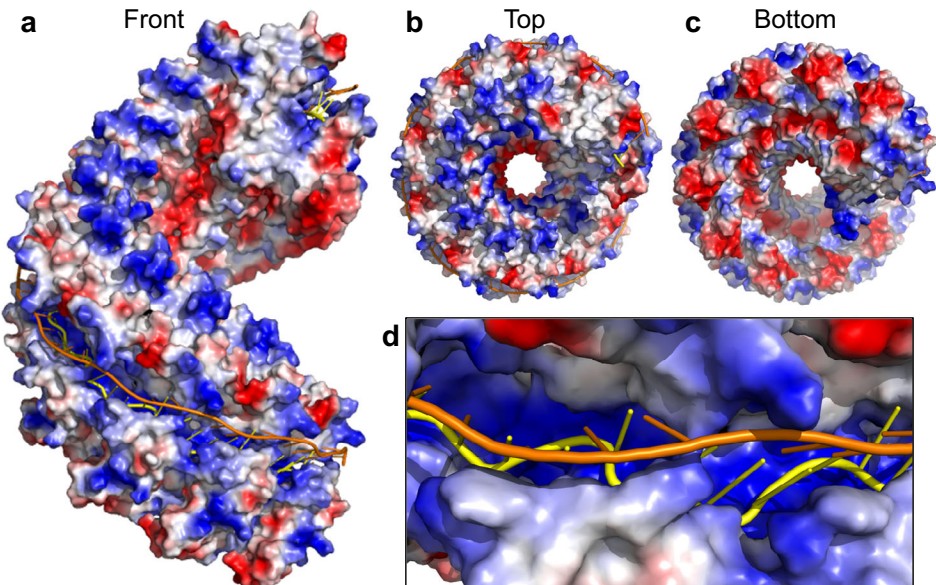

**Fig. 2 | Electrostatic surface views of the LiRecT filament.** Surface colors correspond to regions of positive (blue), neutral (white), and negative (red) charge. The inner DNA strand is yellow and the outer strand is orange. **a** Front view showing the DNA bound to outer positively charged groove. **b** Top view showing the positively charged upper surface of the filament. **c** Bottom view showing the negatively charged lower surface of the filament. **d** Close-up view of the narrow groove showing the strong positive charge where the backbone of the inner strand is bound.

color in yellow is bound to the deepest part of the groove with its nucleotide bases facing outwards. The other strand, which we call "outer" (orange) is bound to the outer portion of the groove with its bases facing inward to form normal Watson-Crick base pairs with the inner strand.

Each monomer of LiRecT binds to 5 bp of DNA. Based on this ratio, we would expect the filament to contain 16–17 subunits of LiRecT bound to the 83-mer duplex. While we do see a filament of approximately this length in the 3D reconstruction, the density towards the ends of the filament gets progressively weaker (Fig. 1c), presumably due to flexibility and/or imperfect alignment of the particles along the filament axis. Consequently, we chose to refine a model that consists of just the 10 subunits of protein and 48 bp of DNA at the central portion of the filament (Fig. 1d), for which the density is strongest. Due to the helical symmetry, however, this model likely encompasses all of the relevant protein-protein and protein-DNA interactions that exist in the full filament, except at the ends. In addition, although the resolution of the map was high enough to clearly see nucleotide bases (Supplementary 2f), purines and pyrimidines could not be distinguished, likely due to the imperfect alignment of the particles along the filament axis. The DNA has thus been modeled as dT48 for the outer strand, and dA48 for the inner strand, despite the fact that both strands contain a natural variation of all four nucleotides. Finally, based on the measured helical parameters of 10 subunits per turn, we would expect the filaments to contain approximately 1.5 turns. In the cryo-EM images, however, many of the filaments contain several turns (Fig. 1a), suggesting that they can stack end-to-end. The result of single particle analysis however converged on just a single 1.5-turn filament.

## LiRecT monomer fold and relation to RAD52
The structure reveals that LiRecT and by extension the RecT/Redβ family of proteins does indeed share structural similarity with RAD52 (Fig. 3), as had been predicted[21,26,27]. In a pairwise superposition using the DALI server[32], the two structures superimpose to an RMSD of 5.5 Å for 83 pairs of Cα atoms that share 14% sequence identity. The structural superposition is shown in Supplementary Fig. 4, and the structure-based sequence alignment in Supplementary Fig. 5. The common core covers 43% of the LiRecT structure of 191 amino acids,

and 31% of the full-length LiRecT protein of 271 amino acids. Despite the common core identified in the pairwise superposition, RAD52 was not identified as a top hit in a DALI search of the Protein Data Bank for structural homologs[32], reflecting the high degree of structural difference. The common core fold consists of 2 central α-helices (α2 and α3) that form the base of the DNA binding groove, combined with a beta hairpin (β1–β2) on one side and a three-stranded antiparallel beta sheet (β3–β5) on the other. In Fig. 3 we have numbered the common core secondary structural elements of LiRecT based on RAD52 and used letters for inserted elements, which are shown in green. The first insertion is an N-terminal 3-helix bundle (αA, αB, αC) that sits at the upper rim of the filament and packs with neighboring copies of itself from the adjacent subunits (Fig. 4). The second is a β-hairpin (βA-βB) inserted after β3 that interacts with β3 of the neighboring subunit at the lower rim of the filament (Fig. 4). The third is a pair of α-helices (αD and αE) inserted after β5 that pack with α3 to form the lower rim of the DNA-binding groove (Fig. 4). Compared to RAD52, the β3–β5 sheet is shorter in LiRecT: the upper portions of β3 and β4 fold back onto the sheet to form the βA–βB insertion, and the upper portion of β5 is replaced by the αD-αE helical hairpin.

The modeled portion of each LiRecT monomer consists of residues 34–224 of the 271 amino acid protein. The additional residues at the N- and C-terminal ends, which are presumably disordered relative to the main body of the filament, would project from the upper and inner surfaces of the filament, respectively (Fig. 4a). Comparisons of the LiRecT structure to predicted structures of it and of λ-Redβ from RoseTTAFold are shown in Supplementary Fig. 6[33]. A structure-based sequence alignment of LiRecT and λ-Redβ is shown in Supplementary Fig. 7.

## Protein−DNA interactions
The two strands of DNA are bound to a deep, narrow, positively charged groove that runs along the outer surface of the filament (Fig. 2). The base of the groove is formed by α2 and α3 and its outer walls are formed by the β1–β2 hairpin on one side and the αD–αE insertion on the other (Figs. 3 and 4). The inner strand is bound to the deepest part of the groove where it is contacted by the side chains of Y110 and K111 from α2, and K206, R210, N211, and K215 from α3

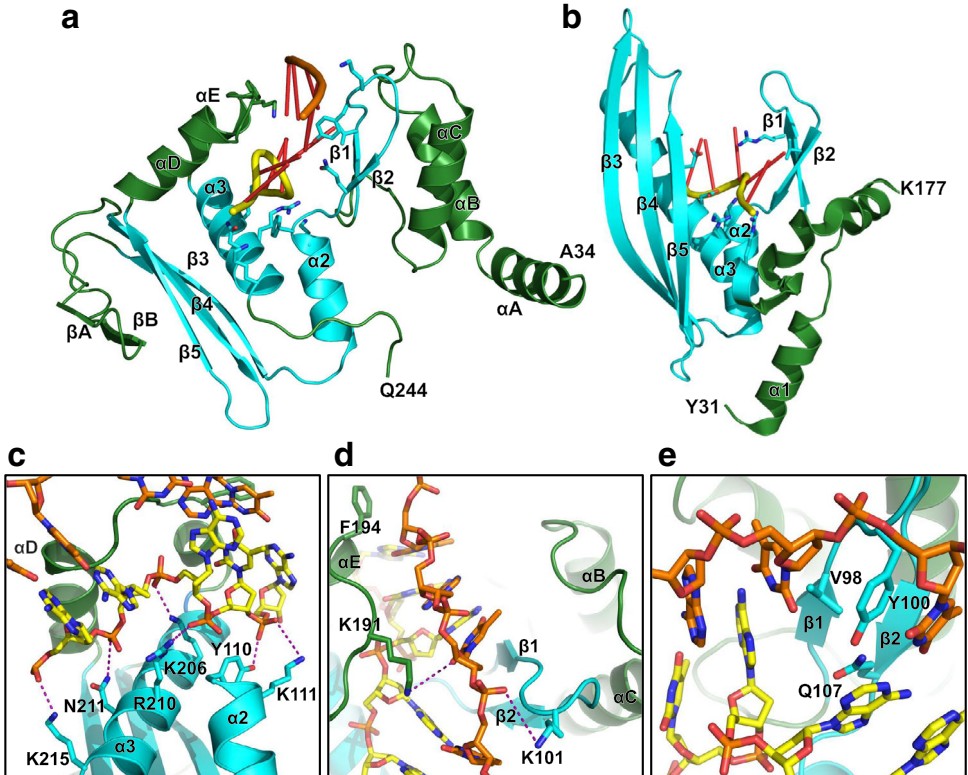

**Fig. 3 | Structure of the LiRecT monomer, comparison with RAD52, and interactions with DNA.** Monomers of LiRecT (**a**) and RAD52 (**b**) are shown in similar orientations with their common core folds in cyan and extraneous segments in green. The DNA backbones are drawn with the inner strand in yellow and the outer strand in orange (for LiRecT only). The DNA binding groove is formed by the 2 central α-helices (α2, α3), the β1−β2 hairpin, and αE-αD (LiRecT) or β3−β5 (RAD52). RAD52 is drawn with coordinates from PDB accession ID 5XRZ[30]. Close-up views of LiRecT interactions with the inner strand (**c**), outer strand (**d**), and β1-β2 hairpin (**e**). Hydrogen bonds within 3.5 Å and ion pairs within 6 Å are shown as dotted lines.

(Fig. 3c). These residues form extensive interactions with the sugar phosphate backbone of the inner strand and hold it in an irregular conformation that is periodically kinked (Fig. 4). By contrast, the outer strand makes far fewer interactions with the protein and adopts a smoother conformation that is held in place primarily by normal Watson−Crick base pair interactions with the inner strand. The few residues that do contact the outer strand are K101 at the tip of the β1−β2 hairpin, and K191 and F194 from the αD−αE insertion (Fig. 3d). Although most of the contacts involve the sugar-phosphate backbone of each strand, the side chains of V98, Y100, and Q107 from the β1−β2 hairpin wedge into the base pairs at every 5th bp step to separate them (Fig. 3e). Specifically, the phenyl ring of Y100 of each subunit stacks with the base of every 5th nucleotide of the outer strand, while the side chain of Q107 contacts the opposing base of the inner strand. These interactions introduce a dramatic kink in the backbone of the inner strand where the bases are separated (Fig. 4). Many of the residues that contact the DNA, particularly those that contact the inner strand, are highly conserved among six distant homologs of LiRecT identified by PSI-BLAST (Supplementary Fig. 8). This suggests that the structure has captured a functionally relevant state of the protein.

Although the two DNA strands mostly contact one another via normal Watson Crick base pair interactions, the duplex is highly extended compared to B-form DNA and completely un-wound (Fig. 5). In concert with the 5 bp/monomer stoichiometry, the bases are stacked in a repeating pattern, with groups of 5 bp stacked with approximately 3.8 Å spacing, alternating with a larger 9 Å spacing where the β1−β2 insertion occurs (Fig. 5c). Overall, the duplex is about 1.5 times as extended as B-form DNA and is completely unwound. The local base pair step parameters deviate significantly from B-form DNA in a regularly repeating manner every 5 bp (Supplementary Fig. 9). This is largely due to the irregular and bent conformation of the inner strand.

## Inter-subunit Packing

The LiRecT subunits pack in the filament with interactions that bury 1830 Å² of total solvent accessible surface area. The interface largely consists of two separate hydrophobic cores, one formed by the N-terminal helix bundles on top of the DNA binding groove, and the other by the β3−β5 sheet and α2−α3 below the DNA binding groove (Fig. 4). The upper core is formed by F41, V44, T76, and T83 from the left subunit (as viewed in Fig. 4b), and F52, L53, L56, and L57 from the right subunit. The lower core is formed by F171, W216, and I218 from the left subunit, and I114, L118, and I126 from the right subunit (Fig. 4c). Both of these cores are surrounded by smaller sets of electrostatic interactions. At the upper rim, K40 and S77 of the left subunit form hydrogen bonds with D46 and N61 of the right subunit (Fig. 4b). At the lower rim, E144 and R141 of the left subunit form hydrogen bonds with N127 and E135 of the right subunit (Fig. 4c). Many of the residues involved in the inter-subunit packing are conserved in distant homologs of LiRecT (Supplementary Fig. 8), suggesting that the subunit packing and overall filament structure are likely to be conserved.

## Comparison to RAD52

The LiRecT structure permits a structure-based sequence alignment with RAD52 to identify the equivalent sets of residues used for interacting with DNA and neighboring subunits (Supplementary Fig. 5). First and foremost, the inner strand in the complex with LiRecT closely overlays with the dT40 bound to the "inner" site of RAD52 (Fig. 3 and Supplementary Fig. 4a). Both strands are bound to the same position deep at the base of their respective grooves, where they are contacted

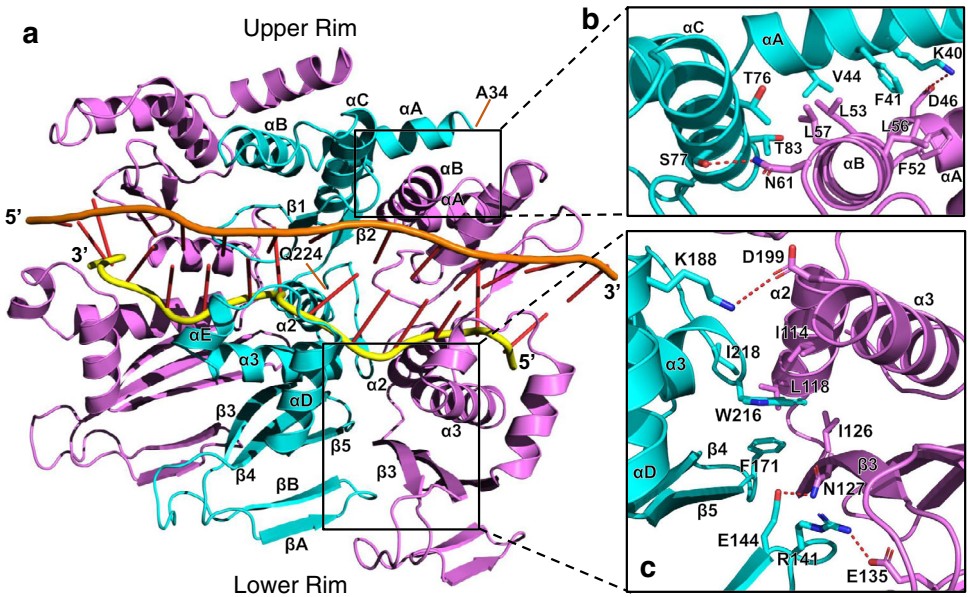

**Fig. 4 | Close-up view of the DNA binding groove and inter-subunit packing.**
**a** Front view of 3 subunits of the LiRecT filament with secondary structures and terminal residues (A23 and Q224) labeled for the middle (cyan) subunit. Close up views of the inter-subunit interactions above (**b**) and below (**c**) the DNA binding groove. Hydrogen bonds within 3.5 Å are shown as dotted lines.

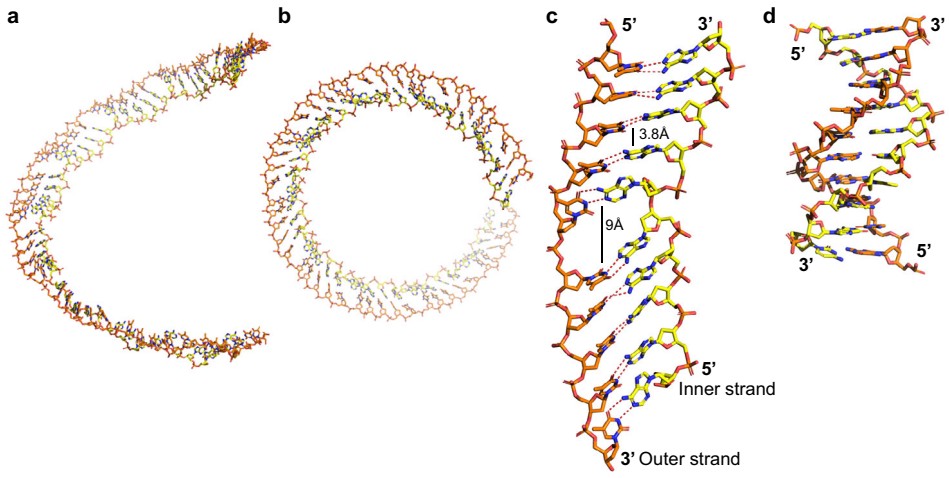

**Fig. 5 | Extended and unwound conformation of the duplex intermediate.** The inner strand is shown in yellow and the outer strand orange. **a** View of 48 bp of the duplex with the filament axis oriented vertically. **b** Top view showing that the inner strand is always closer to the filament axis than the outer strand. **c** Close-up view of a 10 bp segment from the central portion of the filament. The base pairs are spaced by 3.8 Å except at every 5th bp step where they are opened to 9 Å by insertion of the β1-β2 hairpin. **d** 10 bp of B-form DNA drawn to scale for comparison. Coordinates of B-DNA are from PDB code 1BNA[50]. Notice that the duplex intermediate from the LiRecT filament is highly extended and unwound, but still forms normal Watson-Crick base pairs, as indicated by the dotted lines.

by equivalent sets of residues extending from α3 (Supplementary Fig. 10). Specifically, K206, R210, N211, and S214 from α3 of LiRecT correspond precisely to T148, K152, R153, and R156 from α3 of RAD52 (Supplementary Fig. 10c, f). The outer strand of LiRecT approximately overlays with the ssDNA bound to the outer site of RAD52, but the latter is bound in a helical conformation that is clearly not poised for annealing (Supplementary Fig. 10b, d, and g). Both proteins use the conserved β1–β2 hairpin to wedge into the DNA strands, and V98 from β1 of LiRecT is precisely equivalent to R55 from β1 of RAD52. In LiRecT the β1–β2 hairpin separates the base pairs by 9 Å, whereas in RAD52 it separates the inner strand bases by 11 Å (Supplementary Fig. 10e, h). Although our structure of LiRecT captures the protein in a helical filament, and the structures of RAD52 reveal an 11-mer oligomeric ring, the two proteins use the same basic parts of their monomers for inter-

subunit packing (Supplementary Fig. 11), suggesting that the oligomers could be related. At the sequence level, the most conserved part of the LiRecT and RAD52 structures is the interface between α2 and α3, which in both proteins is integral to the binding of the inner strand and the inter-subunit packing interactions. Finally, while the stoichiometry of the RAD52-ssDNA complex is 4 nt/monomer, the complex of LiRecT with annealed duplex has 5 bp/monomer. Whether the two proteins have slightly different stoichiometries, or there is a change in stoichiometry when the second strand is incorporated, remains to be determined.

**Structure of LiRecT in Complex with ssDNA.** Prior work on λ-Redβ revealed that it binds ssDNA as oligomeric rings, and then forms helical filaments once a second complementary strand is added[20]. To

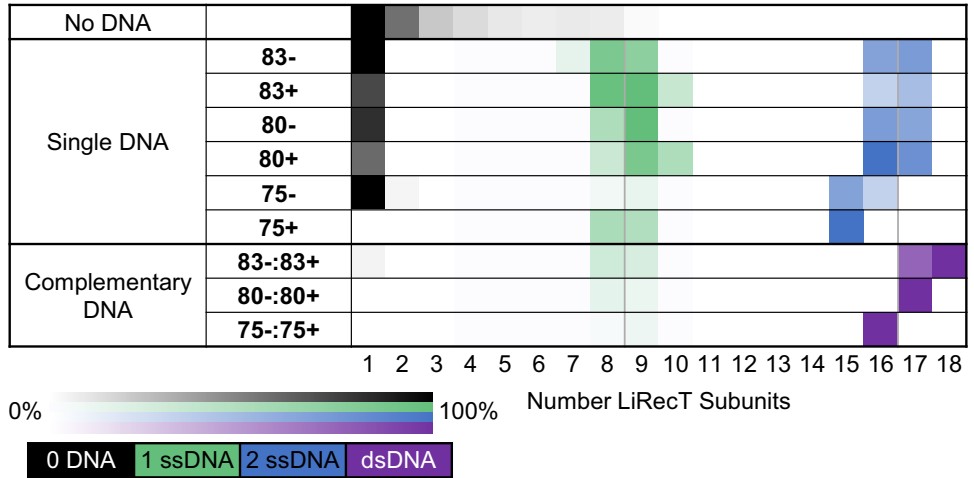

**Fig. 6 | Native MS heat maps of LiRecT oligomers formed in solution.** The first row (No DNA) shows the oligomers formed by 2 μM LiRecT in the absence of DNA. The second set of rows (Single DNA) shows the species formed after mixing a single DNA with 2 μM LiRecT. The third set of rows (Complementary DNA) shows the species formed after mixing two complementary DNAs sequentially with LiRecT. The order in which the DNAs are written corresponds to the order of addition. The heat maps indicate the relative intensities of all species present in each deconvolved spectrum (Supplementary Figs. 15 and 16 and Supplementary Table 2). The coloring corresponds to the DNA present in each complex: black to 0 ssDNA, green to 1 ssDNA, blue to two copies of the same ssDNA (2 ssDNA), and purple to one copy each of two complementary ssDNAs (dsDNA). The data are provided in Supplementary Table 2.

---

determine if there is a similar structural transition for LiRecT, we prepared a complex of it with just one 83-mer ssDNA and obtained a ~5 Å resolution cryo-EM reconstruction by single particle analysis (Supplementary Figs. 12 and 13). Surprisingly, the LiRecT-ssDNA complex also exists as left-handed helical filaments, instead of as rings, but they are not as well ordered, and they do not stack end-to-end. Using a monomer of LiRecT from the complex with annealed duplex, 8 subunits of LiRecT could be docked into the reconstruction for the central portion of the filament. Due to the lower resolution of this reconstruction, we could not fit the ssDNA to the map, although there is strong density for DNA in the groove above α2 and α3 (Supplementary Fig. 12f). Moreover, cryo-EM images collected for LiRecT protein without DNA reveal smaller particles that are much less well ordered (Supplementary Fig. 14), suggesting that the filament is assembled on the ssDNA (a full data set without DNA was not collected). Strikingly, the density for the portion of each LiRecT subunit at the upper rim of the filament is almost completely absent in the structure with ssDNA, for the entire length of the filament (Supplementary Fig. 12). This upper N-terminal lobe of each monomer (N-lobe), which is formed by the αA–αC bundle and the β1–β2 hairpin (Fig. 4a), would likely clamp down on the DNA once the second strand is incorporated, to form the additional protein-DNA and inter-subunit interactions that are shown in Fig. 4 for the complex with annealed duplex. These interactions would further stabilize the filament complex to consolidate annealing. This provides a possible structural explanation for the dramatic increase in stability of the complex with two complementary strands that has been observed by gel-shift and single-molecule experiments for λ-Redβ[19,34].

### Analysis of LiRecT-DNA complexes formed in solution

To determine if the complexes of LiRecT seen by cryo-EM also exist in solution, and in particular if the predicted full-length complex with two 83-mers is formed since the ends of the filament were less well-ordered, mixtures of LiRecT protein alone, with 83-mer ssDNA, and with two complementary 83-mers added sequentially were analyzed by native mass spectrometry (nMS). Raw and deconvolved mass spectra for each sample are shown in Supplementary Figs. 15 and 16, and a heat map summary of the oligomeric species formed for each protein-DNA mixture is shown in Fig. 6. The data used to generate Fig. 6 are shown in

Supplementary Table 2. Free LiRecT protein was largely monomeric at low concentration (1 μM), and while increasing the concentration to 30 μM resulted in some oligomer formation (up to 9-mers), no distinct oligomeric species was converged upon (Fig. 6 and Supplementary Fig. 15). This supports the conclusion from cryo-EM images that filament assembly requires ssDNA. Mixing of LiRecT with one 83-mer ssDNA resulted in two types of complexes, one with 7–10 LiRecT subunits and one copy of the 83-mer (green in Fig. 6), and another with 15–17 subunits of LiRecT and two copies of the same 83-mer (blue in Fig. 6). Based on our previous results for λ-Redβ[31], we interpret the smaller complexes (green) as initial LiRecT-ssDNA substrate complexes, and the larger complexes (blue) as attempts at annealing at sites of partial complementarity. By contrast, mixing of LiRecT with the two complementary 83-mers added sequentially resulted in a more dominant complex containing 17 or 18 copies of LiRecT and one copy each of the 83+ and 83− oligonucleotides (purple in Fig. 6). The stoichiometry of the complex observed by nMS (83/17 or 83/18) is 4.9 or 4.6 bp/monomer, very close to the 5 bp/monomer observed for the cryo-EM structure. Moreover, complexes of LiRecT formed on slightly shorter pairs of complementary oligonucleotides (80- and 75-mers), contained 1–2 fewer subunits, as expected for a continuous oligomerization process like that of a helical filament. By contrast, the complexes of LiRecT with just one ssDNA (green in Fig. 6) did not get noticeably smaller on the shorter ssDNAs, suggesting a different type of oligomerization process for the ssDNA complex. Whether the cryo-EM structure of the 83- ssDNA complex shown in Supplementary Fig. 12 has captured the complexes with one copy of ssDNA seen by nMS (green in Fig. 6) or the complexes with two copies of ssDNA (blue) is uncertain. Based on their apparent length in the 2D class averages (Supplementary Fig. 12b) it is likely to be the latter.

### Mutational Analysis

To test the functional significance of the interactions observed in the structure selected residues were mutated to alanine (or other amino acid types) and the effects on DNA binding and annealing were determined. A total of 42 mutations were targeted to 21 residues forming key interactions at four different regions of the structure (Fig. 7): the inner strand (W96, Y110, K111, H185, K206, R210, N211, and K215), the outer strand (K101, K191, and F194), the β1–β2 hairpin that wedges into both

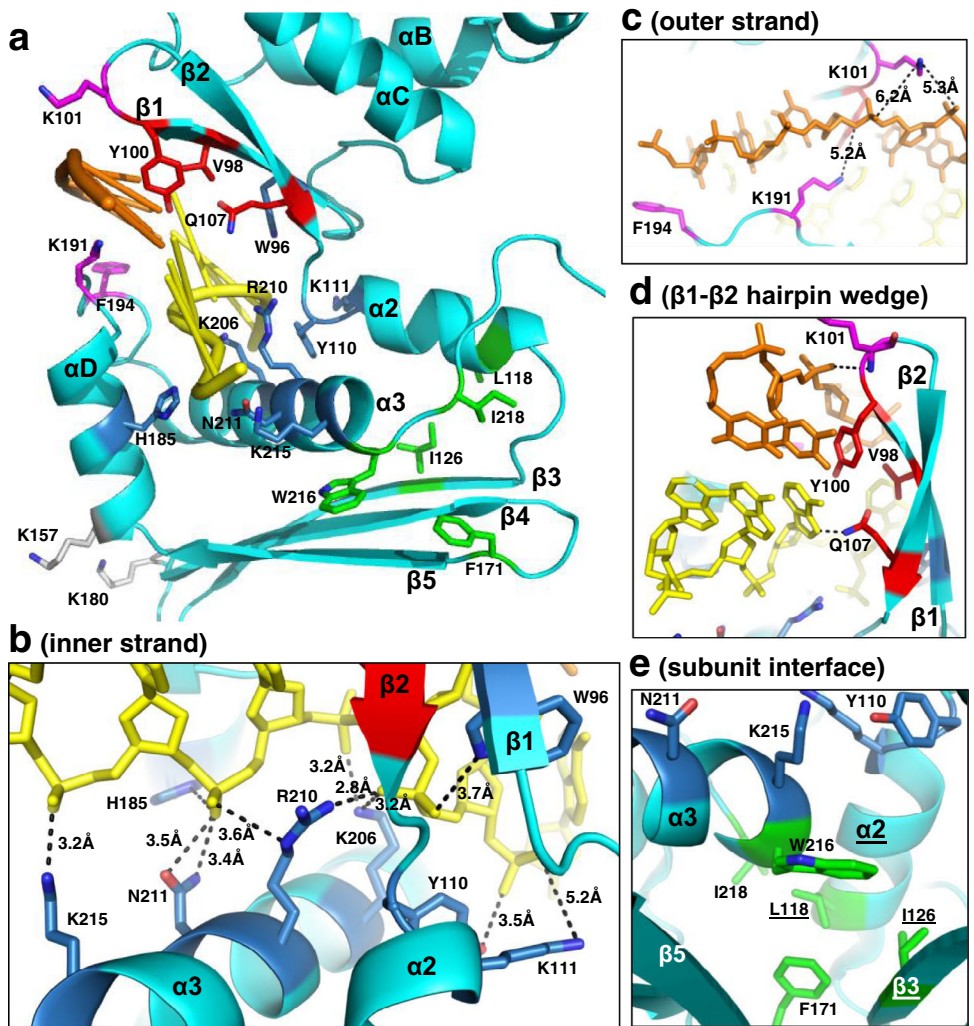

**Fig. 7 | Residues of LiRecT targeted for mutational analysis. a** View of an LiRecT monomer with residues color-coded by location in the structure: blue for inner-strand binding, magenta for outer strand binding, red for β1–β2 hairpin wedge, green for subunit-interface, and gray for control (no interactions). **b**–**e** Close-up views of the four different regions. Hydrogen bonds and ion pairs are shown in dashed lines with distances indicated in Å. Notice that the interactions with the inner strand (yellow) in panel b are much more extensive than those with the outer strand (orange) in panels (**c**, **d**). In panel **e** the labels for the residues and secondary structures of the right subunit are underlined.

strands (V98, Y100, and Q107), and the subunit interface (L118, I126, F171, W216, and I218). As controls, two of the 42 mutations (K157A, K180A) were introduced at surface-exposed residues that are distant from the DNA binding groove and make no interactions in the structure. Three of the protein variants could not be purified, presumably because they disrupted folding and/or solubility. All of these were at the subunit interface (I126H, W216R, and L118A/F171A). The other 39 protein variants could be purified and concentrated (Supplementary Fig. 17), consistent with their being properly folded and soluble.

A gel-shift assay (Fig. 8) was used to test the ability of each variant to bind to 50-mer ssDNA (Cy3-50mer or Cy5-50mer, lanes labeled "3" or "5") and form the complex with duplex intermediate when the two complementary 50-mers are added sequentially (lanes labeled "35"). Although only one experiment for each variant is shown in Fig. 8, all of the experiments were performed multiple times (at least twice), with very similar results. As expected, the two negative control mutations had little to no effect (Fig. 8a, lanes next to WT). For the eight inner strand residues (Fig. 8a), only one of the single alanine mutants (K111A) noticeably disrupted DNA binding. This may be due to the large network of interactions involved in the interaction, such that truncation of only one interacting side chain has minimal effect. Therefore, three

charge reversal mutations (K206E, R210E, and K215E) and four double mutations (K206A/K215A, K206A/R210A, K111A/K215A, and R210A/K215A) of the four positively charged residues were tested. Indeed, all of the double mutations, and one of the charge reversals (K206E) resulted in little to no detectable DNA binding under the conditions tested.

Mutations were also introduced at the three residues that contact the outer strand: K101, K191, and F194. The contacts formed with the outer strand are in general much less extensive and more distant than those formed with the inner strand (compare Fig. 7b and Fig. 7c), and this is reflected in the mutational analysis (Fig. 8b). The mutations included alanine mutants (K101A, K191A, and F194A), charge reversals or insertions (K101E, K191E, and F194E), and one double mutant (K191A/F194A). Only one of these mutations, K191A/F194A, slightly disrupted binding to the duplex intermediate (lane labeled "35"). Overall, the lack of strong effects of the outer strand mutations is consistent with the lack of strong interactions formed by these residues in the structure (Fig. 7c), and with their general lack of conservation in distant RecT/Redβ homologs (Supplementary Fig. 8).

Mutations were also introduced at three residues of the β1–β2 hairpin that wedge into the bases of the duplex to separate them: V98,

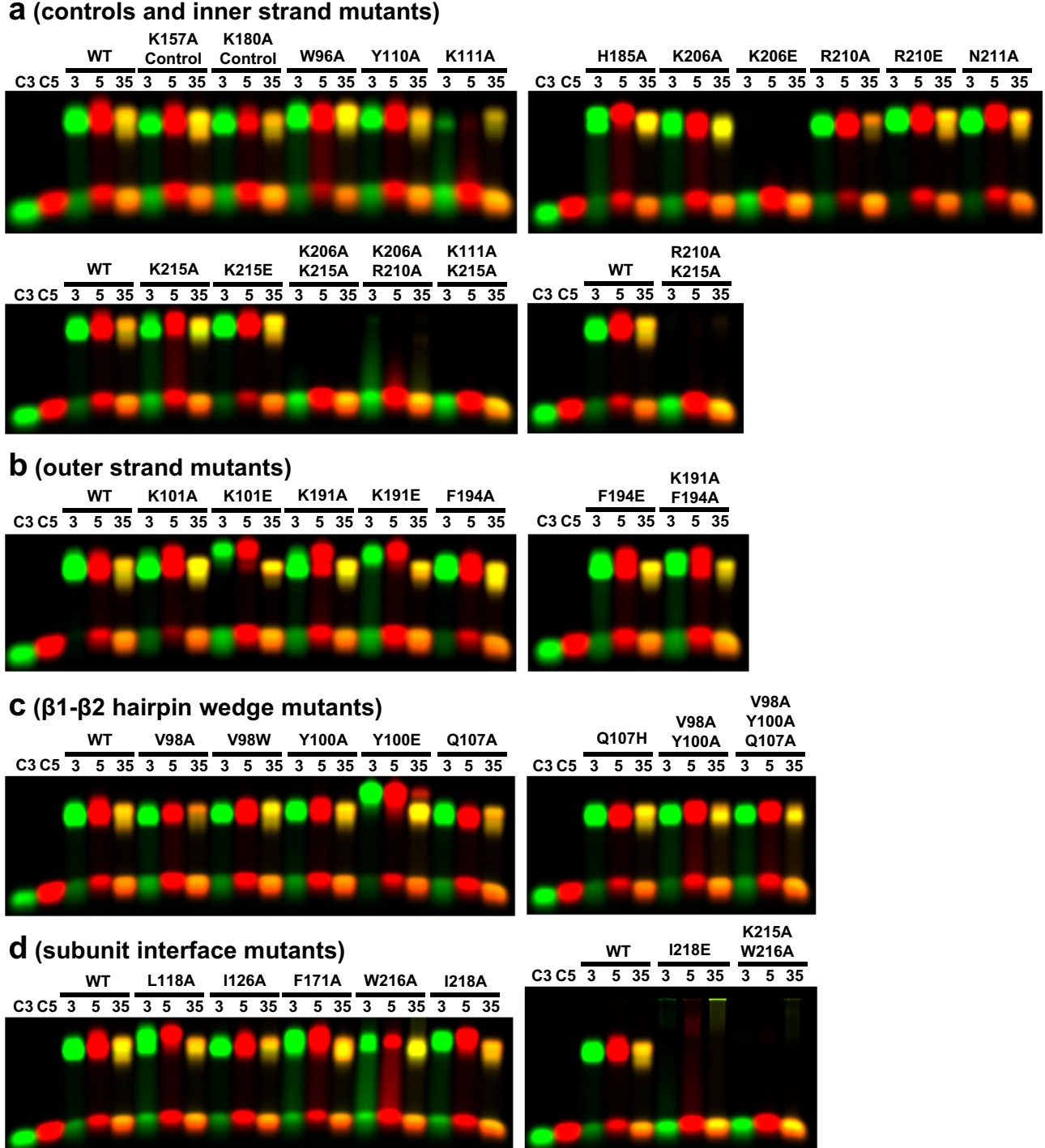

**Fig. 8 | Mutational analysis.** Each panel shows a gel-shift assay with 3.6 µM of LiRecT mixed with different combinations of Cy3- and Cy5-labeled 50mer oligonucleotides (25 µM nucleotides). Lanes C3, C5: each oligo without protein. Lanes 3, 5: LiRecT mixed with each individual oligonucleotide (Cy3-50mer or Cy5-50 mer) to form a ssDNA complex. Lanes 35: LiRecT incubated with Cy3-50mer at 37° for 15 min, followed by addition of Cy5-50mer and incubation for an additional 15 min to form the duplex intermediate (yellow band). Each row contains two images from the same gel (top and bottom halves which have up to 20 lanes each), but are shown side by side to save space for presentation. Panels **a**–**d** show the results of 39 different variants grouped according to their location in the structure, as shown structurally in Fig. 7. At least one WT is shown for each pair of gels to allow for direct comparison. All gel images were obtained with the same scanning conditions and no further adjustments to contrast. Although only one gel is shown for each group of protein variants, each experiment was performed multiple times (at least twice), with very similar results. Source data are provided as a Source Data File.

Y100, and Q107 (Figs. 7d and 8c). Six single mutations (V98A, V98W, Y100A, Y100E, Q107A, and Q107H) had minimal effects, although V98A did have a noticeable reduction in binding to duplex intermediate (lanes labeled "35") as compared to ssDNA (lanes labeled "3" and "5").

Such behavior, a specific defect for binding duplex intermediate (with normal binding to ssDNA), would be expected for a wedge mutation. However, double (V98A/Y100A) and triple mutants (V98A/Y100A/ Q107A) were purified and characterized, and had minimal effects.

V98 is conserved as Val, Ile, or Leu in distant homologs (Supplementary Fig. 8), and it may be that mutation to Ala or Trp does not alter the interaction enough to have a significant effect. Y100 on the other hand is conserved as Tyr or Phe and stacks against the outer strand base of every fifth nucleotide (Fig. 7d). The mild effects of the Y100A mutation are thus surprising. Q107 makes a more subtle interaction with the inner strand base and is not conserved in distant homologs. Interestingly, a relatively close contact (3.5 Å) is formed between the backbone amide of K101 at the tip of the β1–β2 hairpin and a backbone phosphate of the outer strand in this region of the structure (top of Fig. 7d). It may be that the β1–β2 hairpin secondary structure element as a whole, as opposed to specific side chain interactions, is important for the proposed function in clamping down on the duplex to consolidate annealing. If the outer strand is not drawn in fully via complementary base pair interactions, the duplex would likely be too wide to allow the β1–β2-hairpin to clamp down on it.

Finally, mutations were introduced to five apolar residues that make contacts at the lower portion of inter-subunit interface: L118, I126, F171, W216, and I218 (Figs. 7e and 8d). As mentioned above, four of the mutations to the subunit interface disrupted folding and/or solubility, suggesting that this region of the structure is particularly sensitive to mutation (all of these were either charge insertions or double mutations). For those variants that could be purified, single mutations to alanine generally had minimal effects on DNA binding. However, the introduction of a negative charge at the interface in the form of the I128E mutation disrupted DNA binding almost completely. A small amount of aggregates that stayed in the gel well were seen for the complex of I128E with duplex intermediate (lane labeled "35"). The W216A mutation was also combined with mutation of a neighboring residue that contacts the inner DNA strand in the K215A/W216A double mutant (Fig. 7e). This disrupted DNA binding completely.

Collectively, the mutational analysis generally supports the interactions seen in the structure, but due to the large network of interactions involved, particularly at the inner strand and the subunit interface, stronger mutations such as charge reversals or double-mutations are generally needed to disrupt DNA binding. Some of the mutations, in particular Y100E, K101E, and K191E, reduced the migration of the ssDNA complexes, or split them into two bands (K191A). While this could conceivably be due to the effects of the extra negative charge on electrophoresis, other charge reversal mutants (R210E, K215E) did not exhibit this behavior, and some mutations not affecting charge (L118A) did. Conceivably, the differences in mobility, which were most evident for the ssDNA complexes, could be related to the two different-sized complexes that were seen by nMS (green and blue in Fig. 6).

Given the nature of the single-strand annealing reaction, where ssDNA-binding and annealing can be biochemically separated, it should in principle be possible to design mutations that specifically disrupt the formation of the complex with duplex intermediate, without disrupting the complex with ssDNA substrate. Two of the mutants, K191A/F194A and V98A, show signs of this behavior, although additional mutations and more quantitative analysis will be needed to confirm this. Our subunit interface mutations were targeted to the core of the interface underneath the DNA-binding grove, at the C-terminal lobes of the LiRecT monomers, as this forms the bulk of the interface. Future experiments will target the N-terminal lobes above the DNA-binding groove, which are mobile in the complex with ssDNA but clamp down on the duplex when the complementary strand is bound.

## Discussion

Using gel-shift assays with 33-mer and 83-mer oligonucleotides, Radding and colleagues discovered over 20 years ago that λ-Redβ exhibits unusual DNA binding properties: it binds weakly to ssDNA, not at all to pre-formed dsDNA, but tightly to a duplex intermediate of annealing formed when two complementary oligonucleotides are added to the protein sequentially[19]. They referred to this complex as an "intermediate" of annealing, rather than as a "product", presumably because the DNA remained tightly bound to the protein. These experiments did not inform on the conformation of the bound DNA, and whether it was close to B-form or adopted some other conformation remained unknown. Our structure of LiRecT now reveals that the conformation is indeed quite distinct from B-form in being highly extended and completely unwound. Exactly where this conformation of DNA duplex lies along the energetic landscape of protein-mediated annealing (i.e., if it is a transition state or an intermediate), and whether or not it is a special conformation of DNA that is fundamental to annealing and common to all RecT/Redβ family members remains to be determined.

Shortly after the unique DNA binding properties of λ-Redβ were discovered oligomeric structures of λ-Redβ were visualized that closely paralleled the different DNA-bound states: rings for binding to ssDNA and helical filaments for binding to annealed duplex[20]. The filaments of LiRecT that we have observed by cryo-EM closely match the filaments of λ-Redβ seen by negative stain EM: they are left-handed, and have similar dimensions and helical parameters. Given that LiRecT and λ-Redβ share limited sequence identity with one another (<15%), the fact that they share a conserved helical filament structure would tend to suggest that the conformation of the duplex intermediate that is bound to them is also conserved.

Egelman and colleagues predicted that the duplex intermediate formed by λ-Redβ was likely to be bound along the inner surface of the helical filament (though not along the helical axis), based on the observation that it was protected from DNAse I cleavage[19,20]. Based on data from atomic force microscopy and geometric considerations, Stewart and colleagues proposed an alternative model in which an extended and un-wound DNA duplex spirals around the surface of the protein filament to form a right-handed helix[21]. The duplex intermediate bound to our structure of LiRecT is also fully un-wound, but binds to a groove that remains on the outer surface of the filament. The fact that the DNA is buried in such a deep and narrow groove, and that its conformation is far from B-form, may explain why it is protected from DNAse I cleavage.

We have so far not been able to visualize oligomeric rings of LiRecT bound to ssDNA, like the 11-mer rings seen for λ-Redβ[20] and RAD52[28–30]. Our nMS data indicate that LiRecT exists in a monomer-oligomer equilibrium (up to 9-mer) in the absence of DNA, and as a complex of 7 to 10 subunits on a single 83-mer ssDNA. Interestingly, in the complexes with a single 83-mer ssDNA, LiRecT does not appear to bind along the full length of the DNA, as it does for the complex with annealed duplex. These observations are similar to our previous nMS analysis of λ-Redβ[31], although the latter protein had a higher propensity to form oligomers in the absence of DNA. We favor a model in which RecT/Redβ proteins oligomerize weakly and dynamically on their own, assemble onto ssDNA as clusters of cooperatively bound monomers to form partial rings or filaments, and form more stable helical filaments once the complementary strand is incorporated. The weaker complexes on ssDNA may allow for dynamic sampling with multiple strands of ssDNA until a complementary sequence is found and aligned, at which point the N-terminal lobe of each protein monomer likely clamps down on the duplex to stabilize the complex and consolidate annealing.

Filaments of both λ-Redβ and LiRecT can be several helical turns in length, but annealing assays with λ-Redβ indicate that the minimal length needed for successful annealing in vitro is only 20 bp[21,31]. Moreover, oligonucleotides as short as 35-mers are routinely functional for Redβ annealing in vivo[11]. Therefore, we consider it unlikely that long helical filaments of these proteins would form in vivo. Although the helical filament is highly stable, at least as compared to the ssDNA complexes[19,34], it would likely disassemble in vivo once the two DNA molecules are spliced together, possibly due to the greater torsional stress of being bound to the middle of a larger DNA duplex,

as opposed to at the ends. An alternative possibility is that a DNA helicase or a component of the DNA replication machinery could be involved in removing the protein from the DNA in vivo.

While our LiRecT cryo-EM structure captures what appears to be an important intermediate of DNA annealing, cellular DNA annealing reactions likely involve interactions with partner proteins. λ-Redβ forms an interaction with λ exonuclease, which resects dsDNA ends to form the 3′-overhang[35]. This interaction presumably loads the annealing protein directly onto the 3′-overhang as it is being formed, before it can fold into secondary structures. λ-Redβ also forms an interaction with the host single-stranded DNA binding protein (SSB)[35]. This interaction presumably directs the initial λ-Redβ-ssDNA complex to the lagging strand of the replication fork, where it can pair with the complementary target site as it is exposed. Such coordinated interactions are likely to be shared by other RecT/Redβ family annealing proteins, including LiRecT.

Residues 1–33 and 225–271 of LiRecT were not resolved in our 3D-reconstruction of the filament. These residues are however part of a RoseTTAFold model for the LiRecT monomer, as shown in magenta in Supplementary Fig. 6b. Residues 1–33 form two α-helices, one that is quite long (residues 1–27) and extends away from the core of the monomer, and another that is short (residues 27–33) and forms a right angle with αA. In our reconstruction, there is density for what appears to be a helix preceding αA. Although the density for this helix was not clear enough to model, it appears to pack against αA of the neighboring subunit, and thereby add to the inter-subunit contacts. There is no sign of density that would correspond to the long N-terminal α-helix from the RoseTTAFold model.

By analogy with λ-Redβ, it is likely that the extra residues at the C-terminal end (225–271) fold into a small helical domain for forming interactions with partner proteins, including the host single-stranded DNA-binding protein (SSB)[35]. In the RoseTTAFold model, residues 242–271 extend away from the filament to possibly form such a domain, but residues 220–238 form an α-helix that packs against the β1–β2 hairpin and would overlap with the DNA if it were bound. The placement of this helix is not consistent with DNA binding, but it could conceivably adopt this position in the LiRecT monomers before they assemble onto the DNA. Further studies will be needed to resolve these issues.

The structure confirms that the RecT/Redβ family of annealing proteins share a common core fold with RAD52. The two proteins use this fold to bind to the first ssDNA in similar ways, with equivalent sets of residues contacting the DNA from common secondary structural elements (α2 and α3). Moreover, the proteins use approximately the same portions of their monomers for inter-subunit packing, suggesting that their oligomers could be related. However, RAD52 has so far only been observed to form rings, and has not been seen to form helical filaments. RAD52 also exhibits somewhat different DNA-binding properties from λ-Redβ in binding with higher affinity to ssDNA and to pre-formed dsDNA[36]. Furthermore, a distinct complex of RAD52 with a duplex intermediate of annealing like those of λ-Redβ and LiRecT has not yet been observed. Nonetheless, the DNA binding grooves on the LiRecT and RAD52 structures are formed by a common set of secondary structural elements, and are similarly deep and narrow, suggesting that a complex of RAD52 with two strands of complementary DNA bound simultaneously could very well be formed. The existence of such a complex would favor a *cis* mechanism of annealing in which the two DNA strands are bound to the same protein oligomer as they are annealed to one another, as opposed to a *trans* mechanism in which annealing is mediated by the interaction of two separate RAD52-ssDNA complexes.

Although human RAD52 has been widely considered to exist as stable oligomeric rings, yeast RAD52 is expressed at only nanomolar concentrations in vivo[37], and human RAD52 is largely monomeric at sub-micromolar concentrations in vitro[38]. Thus, non-ring forms of RAD52 could still be relevant to its mechanism.

Some features of the LiRecT-DNA complex are remarkably similar to other types of DNA recombination proteins. The 1.5× extended conformation of DNA and the 5 bp repeating pattern of extension are similar to the triplet-repeating conformation of DNA bound to *E. coli* RecA protein[39]. The LiRecT-DNA complex also shares some remarkably similar features with a multi-subunit complex of *E. coli* Cascade bound to an RNA-DNA duplex hybrid[40]. In Cascade, the duplex is bound to a very similar groove along the outer surface of a right-handed helical assembly of subunits. The duplex is similarly extended and un-wound, bound in a pattern that repeats every 6 bp steps due to a similar β-hairpin insertion, and has the first strand added (RNA) in the deepest part of the groove and the second strand added (DNA) at the outer part of the groove. These similarities of LiRecT with functionally (but not structurally) related proteins point to fundamental principles of DNA transactions that are still being unraveled.

While our manuscript was in revision, the cryo-EM structure of an N-terminal fragment of λ-Redβ (residues 1–177) corresponding to its DNA-binding domain was reported in this journal[41]. The complex was formed with complementary 27-mer oligonucleotides and adopted continuously stacked left-handed helical filaments. The filaments are much more loosely wound than the LiRecT filaments reported here, as there are 27 subunits per helical turn instead of 10. However, the dimensions of the outer DNA-binding groove and the conformation of the bound DNA duplex are very similar. The fact that two distantly related proteins bind to such a similar conformation of duplex intermediate supports the fundamental importance of the structures to understanding the mechanism of protein-mediated DNA annealing.

## Methods

### Materials

The vendors and catalog numbers for chemicals and other materials used in this study are shown in Supplementary Table 3. All oligonucleotides used in this study were purchased HPLC-purified from Integrated DNA Technologies, dissolved in ddH$_2$O, and stored at −20 °C. Their full sequences are shown in Supplementary Table 4.

### Protein expression and purification

The gene expressing LiRecT (UniProtKB – Q92FL9) was PCR amplified from *Listeria innocua* CLIP 11262 genomic DNA (ATCC BAA-680) and cloned into pET28b between the *Nde*I and *BamH*I restriction sites to express a protein with an N-terminal 6His-tag and a site for thrombin cleavage. The protein was expressed in BL21(AI) *E. coli* cells (Invitrogen) in 6 × 1 L cultures at 37 °C, grown to an optical density at 600 nm of 0.65, and induced by 1 mM IPTG and 0.2% arabinose. At four hours post-induction, the cells were harvested by centrifugation, resuspended in 60 ml of Buffer A (50 mM NaH$_2$PO$_4$, 300 mM NaCl, 10 mM imidazole, pH 8.0) and frozen at −80 °C. After thawing, lysozyme (1 mg/ml), PMSF (0.1 mg/ml), leupeptin and pepstatin (1 µg/ml each) were added and incubated for 60 min on ice. The cells were then sonicated on ice, centrifuged at 38,000 × $g$ for 3 × 30 min, and the final supernatant was loaded on to a 2 × 5 ml HisTrap Fast Flow column (Cytiva) at 0.5 ml/min. The column was washed with 30 ml of Buffer A, 200 ml of Buffer A containing 30 mM imidazole, and eluted with a 200 ml gradient of 30–500 mM imidazole in Buffer A. After SDS-PAGE analysis, pooled fractions were mixed with 100 units of Thrombin (Cytiva), dialyzed at room temperature into Buffer B (20 mM NaH$_2$PO$_4$, 1500 mM NaCl, pH 7.4), and loaded back onto the HisTrap FF column. The flow through was collected, dialyzed at 4 °C into Buffer C (20 mM Tris pH 8.0) and loaded onto a 2 × 5 ml HiTrap Q FF column (Cytiva) at 1 ml/min. After washing with Buffer C for 30 ml, the protein was eluted with a 100 ml gradient to Buffer C plus 1 M NaCl. Pooled fractions were dialyzed into Buffer D (20 mM Tris, 1 mM DTT, pH 8.0), concentrated to 50 mg/ml (Vivaspin 20, 10 kDa MWCO), and stored at −80 °C in 50 µl aliquots. Protein concentration was determined by O.D. at 280 nm using an extinction coefficient of 43,890 M$^{-1}$ cm$^{-1}$, which was

determined from the amino acid sequence, which has 5 tryptophan residues.

## DNA binding assay

A gel shift DNA binding assay used two complementary 50-mer oligonucleotides labeled at the 5′-end with either Cy3 or Cy5. The indicated concentration (5 or 3.6 μM) of Redβ or LiRecT in PBS (or cryo-EM buffer defined below) was mixed with 25 μM (nt) of the indicated oligonucleotide and incubated at 37 °C for 15 min. For some samples as indicated on the gel (lanes labeled "35", "ad" or "nc"), a second oligonucleotide was added and incubated for an additional 15 min at 37 °C. For all samples the total reaction volume was 30 μl. For visualization 17.5 μl of each complex was mixed with 7.5 μl Orange G dye (65% w/v sucrose, 10 mM Tris-HCl pH 7.5, 10 mM EDTA, 0.3% Orange G powder from Sigma Life Sciences), loaded onto a 0.8% agarose gel and electrophoresed in 1× TBE at room temperature for 72 min at 96 V. Gels were imaged using a Sapphire Biomolecular Imager (Azure Biosystems) with Sapphire Capture Software (version 1.12.0921.0). Scanning parameters for Fig. 8 were pixel size 100 μm, scan speed high, 2.38 mm focus, intensity 2 for Cy5, intensity 4 for Cy3, black lighting 50, white 37186, gamma 1.37. Scanning parameters for Supplementary Fig. 1a, b were intensity 1 for Cy5, intensity 2 for Cy3, black lighting 50, white 15362, gamma 0.88.

## Cryo-EM sample preparation

The complex of LiRecT with annealed duplex was prepared by first incubating 0.7 mg/ml protein with one 83-mer oligonucleotide (83−) at a ratio of 4 nt/monomer (94 μM nucleotides) in 20 mM $KH_2PO_4$, 10 mM $MgCl_2$ pH 6.0 at 37 °C for 15 min. Then an equivalent amount of the complementary 83-mer (83+) was added and incubated for an additional 15 min at 37 °C, after which the prepared complex was kept on ice for approximately 90 min. The complex of LiRecT with ssDNA was prepared in the same manner as that for annealed duplex but only the first strand of ssDNA (83−) was added. For both complexes, the total reaction volume was 19 μl. Just prior to vitrification, 1 μl of 1.5 mM n-dodecyl-β-D-maltopyranoside (Anatrace; final concentration at 0.5 CMC) was added and incubated for 30 s, and then 4 μl of the mixture was added to a Quantifoil R1.2/1.3 Au 300 mesh grid (Electron Microscopy Sciences) that had been glow discharged for 60 s at 20 mA using a Pelco easiGlow. After applying the sample, the grid was immediately frozen by plunging into liquid ethane using a Vitrobot Mark IV (Thermo Fisher Scientific) at 4 °C, 100% humidity, 1.5 s blot time, and 0 blot force. Ted Pella 595 filter paper (product # 47000-100) was used for blotting.

## Cryo-EM data acquisition

For the complex with 83-mer annealed duplex, images were collected on a 300 keV Titan Krios G3i electron microscrope (Thermo Fisher Scientific) operating in nanoprobe EFTEM mode with 50 μm C2 aperture, 100 μm objective aperture, a Gatan BioContinuum energy filter (20 eV slit width, zero energy loss), a Cs corrector, and a Gatan K3 direct electron detector operating in counting mode. Automated data collection was performed in EPU with defocus values ranging from −1 to −3.5 μm at a magnification of 81,000× and a pixel size of 0.899 Å (non-super-resolution). The dose rate was adjusted to 24.28 e-/Å²/s with an exposure time of 2.7 s split into 36 fractions to achieve a total dose of 66 e-/Å². A total of 2038 movies were collected. For the complex with 83- ssDNA, the same settings were used, except for the following: the data were collected in super-resolution mode such that the pixel size was 0.4495 Å, the does rate was adjusted to 22.80 e-/Å²/s with an exposure time of 2.83 s split into 45 fractions for a total dose of 65 e-/Å², and 1619 movies were collected.

## Cryo-EM data processing

For the data for the complex with annealed duplex, movies were imported into cryoSPARC v2.15.0[42] for single particle analysis. Patch

motion correction was implemented with a 3 Å maximum alignment resolution and a B-factor of 500. Patch CTF estimation was implemented with an amplitude contrast of 0.1. From the motion- and CTF-corrected micrographs, approximately 1000 particles were manually picked and used for one round of 2D classification. Six 2D class averages representing different particle orientations were chosen and used as templates for automated particle picking, which resulted in approximately 1,100,000 particles. Particles were extracted with a box size of 252 Å and put through three rounds of 2D classification to result in 391,275 cleaned particles. The cleaned particles were used to generate three initial models with ab-initio reconstruction, the best of which (271 K particles) was refined in homogenous refinement to yield a 3D reconstruction with an FSC gold standard resolution of 3.41 Å (tight mask), or 4.3 Å (no mask). After polishing, the resulting 3D reconstruction showed clear density for protein backbone, side chains, and two strands of DNA including bases. The data for LiRecT with 83-mer ssDNA were processed in the same manner to result in 180,965 cleaned particles and a final resolution of 4.79 Å. This resolution is likely over-estimated however as the FSC curve was oscillating. The resulting map reveals clear secondary structure feature but very few side chains.

## Model building and refinement

For the complex with annealed duplex, the two un-masked half maps from cryoSPARC were input into the RESOLVE procedure for density modification in PHENIX version 1.20.1–4487[43] which improved the resolution by 0.22 Å from 3.81 Å (FSCref=0.5) to 3.59 Å (FSCref=0.5). A model of one protein monomer containing residues 34–224 (out of 271 total) was built into the central portion of the filament with COOT version 0.8.7[44], and then transformed iteratively into density for nine neighboring subunits using CHIMERA version 1.13.1[45]. Additional subunits towards the ends of the filament were visible in the reconstruction, but not included in the final model, as the density for these regions was progressively weaker. The 3D reconstruction also showed clear density for 48 bp of DNA duplex at the central portion of the filament, which was also built using COOT. Once a 10-subunit filament was built, the NCS operators were determined from the structure using Find NCS in PHENIX, and then used for 10-fold NCS averaging in Resolve, which further increased the resolution to 3.50 Å (FSCref=0.5). The final model consists of 10 protein subunits and 48 bp of DNA. Real space refinement and model validation in PHENIX yielded a final FSC = 0.143 map to model resolution of 3.2 Å. During refinement, 10-fold NCS constraints were applied to the protein monomers, but not to the DNA. Final refinement and model validation statistics are shown in Supplementary Table 1. For the structure with 83-mer ssDNA, the resolution of the reconstruction did not enable the model to be built from scratch as very few side chains were visible, but six LiRecT subunits from the structure with annealed duplex could be auto-fit into density using CHIMERA, and additional subunits could be fit using PHENIX (dock_in_map). The density corresponding to the N-terminal lobes of each monomer (residues 34–109) was weak and these residues of each subunit were deleted from the model. The final model consisting of residues 110–221 of 8 LiRecT subunits was refined in PHENIX by rigid body refinement only. Structural figures were prepared using PyMOL version 2.5[46]. Atomic coordinates and maps have been deposited in PDB and EMDB under accession codes 7UB2 and EMD-26434 for the complex with 83-mer annealed duplex, and 7UBB and EMD-26437 for the complex with 83-mer ssDNA).

## Native mass spectrometry

LiRecT protein was buffer exchanged into 100 mM ammonium acetate pH 7 (unadjusted) using Micro BioSpin P6 spin columns (Bio-Rad Laboratories, Hercules, CA, USA). All ssDNAs were dialyzed into 100 mM ammonium acetate with Pierce 96-well microdialysis devices with 3.5 K MWCO (Thermo Fisher Scientific). For the preparation of

LiRecT-DNA complexes, LiRecT was diluted to the experimental concentrations indicated, and then the first ssDNA was added at the indicated concentration based on nucleotides (nt) per monomer of LiRecT, and incubated at 37 °C for at least 15 min. For complexes with annealed duplex, the second complementary ssDNA was then added and incubated for an additional 15 min. Samples (3–5 μl) were directly loaded into nanoESI emitters that were pulled in-house from borosilicate filament capillaries (OD 1.0 mm, ID 0.78 mm, Sutter Instrument) using a P-97 Flaming/Brown Micropipette Puller (Sutter Instrument). Experiments were performed on a Thermo Scientific Q Exactive Ultra-High Mass Range (UHMR) mass spectrometer from Thermo Fisher that was modified to allow for surface-induced dissociation (SID, not used in this work) similar to a previously described modification[47]. The same instrument settings were used as described previously[47]. Ion activation was necessary for improved transmission and de-adducting of ions to resolve species at higher m/z. For this, in-source trapping (IST) of −10 V and higher energy collisional dissociation (HCD) of 90 V was used for the LiRecT plus DNA mixtures. All data were deconvolved using UniDec V4.4[48]. A range of deconvolution settings was initially surveyed. The settings optimized for LiRecT plus DNA mixtures were the following: 2000 to 16,000 m/z, charge range of 1 to 70, mass range of 10–800 kDa, sample mass every 10 Da, split Gaussian/Lorentzian, peak FWHM 3 or 4 Th, artifact suppression 40, charge smooth width 2.0, point smooth width 2, and native charge offset −20 to 10 or 20. The use of manual mode to assign a fraction of the peaks with charge states was needed to reduce artifacts. The resulting deconvolutions were plotted as relative signal intensities.

## Mutational analysis
Structure-guided mutations were introduced into the pET28b-LiRecT expression plasmid by the QuikChange™ method (Agilent technologies). The protein variants were expressed from BL21-AI cells and purified by a previously described small-batch version of the method described above[49]. Briefly, cells from 50 ml cultures were re-suspended in 3.0 ml of Buffer A and frozen at −80 °C. Cell suspensions were thawed and incubated for 30 min on ice with 1 mg/mL lysozyme, 1 μg/mL leupeptin, 1 μg/mL pepstatin, and 1 mM PMSF. Cells were then sonicated using a micro-tip, clarified by centrifugation at 38,000 × g for 30 min, and 2.1 ml of the soluble supernatant was loaded onto a Qiagen Ni-spin column (Cat. # 31014) that had been pre-wet with 600 μl of Buffer A. The columns were washed four times with 500 μl of Buffer A containing 30 mM imidazole, and eluted four times with a total of 1.8 ml of Buffer A containing 500 mM imidazole (2 times with 200 μl followed by two times with 700 μl). Pooled fractions (1.8 ml total) were buffer exchanged into Buffer B using PD-10 desalting columns (Cytiva, Cat. # 170851-01), concentrated to 1–8 mg/ml using an Amicon Ultra-4 centrifugal filter with 10 kDa MWCO (MilliporeSigma Cat. # UFC8010), and frozen in 50 μl aliquots at −80 °C. The final purified proteins (Supplementary Fig. 17) retain an extra 20 N-terminal amino acids from the expression vector, which had minimal if any effect on DNA binding. DNA binding assays for each variant were performed as described above, where the WT protein used for comparison was purified by the same small-batch method described for the variants.

## Reporting summary
Further information on research design is available in the Nature Portfolio Reporting Summary linked to this article.

# Data availability
The structural coordinates generated in this study have been deposited in the Protein Data Bank under accession code 7UB2 for the complex with 83+/83− annealed duplex and 7UBB for the complex with 83− ssDNA. The volumes generated in this study have been deposited in the EMDB database under accession codes EMD-26434 for the complex with 83−/83+ annealed duplex and EMD-26437 for the complex with

83− ssDNA. The cryo-EM micrographs used in this study have been deposited in the EMPIAR database under accession code EMPIAR-11348 for the complex with 83+/83− annealed duplex and EMPIAR-11353 for the structure with 83− ssDNA. All unique biological materials, including plasmids used for protein expression, are available from the authors upon request. Source data are provided with this paper.

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

## Acknowledgements

This work was funded by grants from the National Science Foundation (MCB-1616105 and MCB-2212951 to C.E.B) and National Institutes of Health (T32GM11829 to B.J.C. and P41GM128577 to V.H.W.). The content is solely the responsibility of the authors and does not necessarily represent the official views of the National Science Foundation or the National Institutes of Health.

## Author contributions

B.J.C., A.S.N., C.F.K., V.H.W., and C.E.B. designed research, B.J.C., A.S.N., C.F.K., and A.M.W. performed the research, B.J.C., A.S.N., C.F.K., A.M.W., V.H.W., and C.E.B. analyzed data, and B.J.C., A.S.N., and C.E.B. wrote the paper.

## Competing interests

The authors declare no competing interests.
