## [Peer Review File · Nature Communications]

Structure of a RecT/Red β family recombinase in complex with a duplex intermediate of DNA annealingREVIEWER COMMENTS

Reviewer #1 (Remarks to the Author):

Single-stranded DNA (ssDNA) annealing (SSA) is involved in many DNA transactions to ensure genome integrity. It has been well documented that bacteriophage λ Red β , the SSA protein, binds weakly to ssDNA, not at all to preformed dsDNA, but has a higher affinity to a duplex intermediate of annealing formed when two complementary strands of ssDNA are added to the protein sequentially. However, no structural insights of such SSA proteins in complex with the intermediate state of annealing duplex DNA have been delineated. Here, the authors used a single particle cryo-EM approach to solve a 3.4 Å structure of a SSA protein LiRecT from prophage of *Listeria innocua* in complex with the annealing duplex DNA. The structural features of the LiRecT-dsDNA intermediate are revealed as follows: (1) LiRecT exhibits a left-handed helical filament with roughly 10 protomers per helical turn, (2) each monomer binds to 5 bp of dsDNA intermediates, (3) the structures of protomer-protomer and protomer-DNA interaction have been visualized and analyzed, and (4) the bound duplex DNA is highly extended and underwound compared to B-form DNA. In addition, the repeating 5 bases cause a large 9 Å spacing between bases. Finally, the cryo-EM results were verified in solution by native mass spectrometry. It is worth noting that, unlike Red β and Rad52 binding ssDNA as oligomeric rings, the author observed that the LiRecT-ssDNA complex exists in a left-handed helical filament structure.

To my knowledge, this is the first structural documentation of the SSA protein with the intermediate state of annealing duplex DNA. Based on the structure information, the authors hypothesize that the N-terminal helix bundles of LiRecT clamp down on the duplex after the second strand is included to stabilize the complex and consolidate annealing. This provides a view of the SSA process at a mechanistic level. In addition, the structural information also revealed the structural similarity between LiRecT and eukaryotic Rad52, suggesting a possible SSA mechanism for Rad52. In summary, the novelty is well presented, and the manuscript is well written. However, there is no functional study to validate their structure observations. Addressing the following concerns would strengthen the evidence presented in the current manuscript:

1. It will be important to validate the structure findings via functional analyses. For example, the side chains of V98, Y100, and Q107 from the β 1- β 2 hairpin wedge into the base pairs at every 5th bp step to separate them (Fig. 2E). It will be interesting to make mutations in those residues to examine the functionality of strand annealing. In addition, given the structural homology with Rad52, it will be appreciated to conduct the functional study in either LiRecT or Rad52. For example, both proteins use the conserved β 1- β 2 hairpin to wedge into the DNA strands, and V98 from β 1 of LiRecT is precisely equivalent to R55 from β 1 of Rad52. Could we examine the effect of those mutant variants on SSA activity?

2. In the cryo-EM structure of the LiRecT-ssDNA complex, the author could not see the ssDNA due to the low resolution of reconstruction. This significantly attenuates the importance of marking the structural differences between Rad52-ssDNA and LiRecT-ssDNA complexes. The authors should prove that a ssDNA indeed exists within the complex. Does LiRecT form a helical filament in the condition without DNA under cryo-EM?

3. Is any reason that the author used low pH 6 in cryo-EM sample preparation (page 19, line 423)? Does pH affect the LiRecT in complex with ssDNA and annealing duplex?

Reviewer #2 (Remarks to the Author):

The paper by Caldwell et al presents substantial progress in the understanding of single strand annealing proteins (SSAPs). SSAPs were originally grouped together largely because they share some biochemical abilities to anneal DNA and also were not RAD51/RecA ATPases. The possibility that the SSAPs were related and form a RAD52 superfamily (named after the most important member) was proposed some time ago. This paper establishes the veracity of this important proposal and furthermore delivers remarkable mechanistic insight that will promote a great deal of progress. This is a remarkably significant paper that will be a milestone in the understanding of the poorly understood but crucial homologous recombination and DNA repair processes mediated by SSAPs. In particular, the details revealed by high resolution of the annealed DNA in the helical filament will motivate the debate surrounding RAD52 mechanism. Furthermore the diverse properties displayed amongst the vast array of prokaryotic SSAPs can now be evaluated with a new focus to consider key shared properties and diversifications. Since some of these prokaryotic SSAPs (lambda Red beta, E.coli RecT) are extremely useful for recombinant DNA and genome engineering, inventive applied progress is likely to emerge.

The manuscript is superbly written at the highest standards. The data treatment is logical and thorough. I have only a few comments aimed at slight improvements.

Line 61, ref 22 is not the right reference and Datta et al, 2008 PNAS 105, 1626 would be better.

Line 66, this sentence will be better as - "The structure of an 11-mer ring form of the DNA-binding domain of Rad52 was determined without DNA (28, 29) and with an oligonucleotide to form a substrate complex (30), but there is no"

Line 74, this sentence will be better as "The filaments are strikingly similar to those seen for lambda-Red beta at low resolution by electron microscopy and atomic force microscopy (20, 21), but our structure"

Line 297, paragraph - the conclusions that the duplex intermediate formed by Red beta must lie on the outside of the helical filament and the DNA must be strongly unwound (not B-form) have been made before (ref 21).

Line 321 - oligonucleotides as short as 35-mers are routinely used for Red beta recombineering in vivo.

The secondary structure labels on Figures 2A,B,C, 3A, SFig 4, SFig 10 are hard to read in places. Figure 2 legend - denote the strand colours.

Reviewer #3 (Remarks to the Author):

The manuscript by Caldwell et al. reports the structure of a strand annealing reaction intermediate catalyzed by the RecT/Redbeta family of phage strand annealing proteins. There have been several structures of such proteins, including structures with a single ssDNA strand, but the structure of Caldwell et al. is the first instance when we see the annealed dsDNA. This structure represents a major advance in our understanding of single-strand annealing (SSA), a DNA repair process present in eukaryotes as well.

The RecT protein forms a left handed filament and binds to the 1st ssDNA as reported before (though some annealing proteins form rings). The complementary strand is held in place mostly through Watson-Crick hydrogen bonds, making annealing highly dependent on complementarity between the two strands. Globally, the DNA is stretched and under-wound compared to B DNA. Locally, however, the DNA is arranged in groups of 5 nucleotides; within each group the bases are well stacked, which is important for the Watson-Crick hydrogen bonds, while between groups there is a large spacing, suggesting complementarity sampling occurs in groups of 5 nts. These provide key mechanistic insights into the how proteins can catalyze SSA. Conceptually, they are similar to how the RecA and CRISPR Cascade proteins determine complementarity between two strands – a striking demonstration of how different proteins/processes can converge onto the same solution.

Structure-determination is solid. The ambiguity in what is happening at the ends of the filament is explained by flexibility there – a perfectly sensible explanation. The ambiguity in assigning the DNA sequence is also expected given that the filaments can stack end-to-end. None of these materially affect any of the conclusions.

Minor comments:

1) line 126: The term “common core” needs to be defined as the structural elements that align. It should also be indicated on Figs. 2A and 2B (or an adjacent figure). It may be easier to first define the RecT secondary structure elements and then do the comparison; this way the common fold would be easier to explain. Also, an rmsd of 4.3 Å for 107 Calphas (56% of RecT structure) would invariably contain spuriously overlapping residues (at least from a visual comparison of Figs 2A and B). The common core, as it is colored in Fig. S4, contains the beta sheet of strands 3/4/5, but in Fig 2B this sheet has an entirely different position, and while it has the same topology, structurally it is in a very different position; its residues should not be included spuriously in the rmsd measurement. Maybe the “common core” can be defined relative to alpha2/3 and beta1/beta2. A superposition of RecT and rad52 would also be very useful, though making it clear may be hard.

2) Fig. 2B legend should state that Rad52 is oriented according to the superposition (if that’s indeed the case).

3) I am not sure that discussion of RoseTTAFold adds anything of value, but I do not feel strongly about this (lines 145, 331, 339). It can just be stated that there is low-res density that could correspond to an N-terminal helix, but it is not interpretable (after all that is the data, and Rosetta is just prediction).

4) Is the DNA “under-wound” (line 171) or “completely unwound” (line 176) ? or do these statements refer to local and global DNA parameters ?

5) line 231: “tightly wound, with approximately 8 subunits per turn as opposed to ~10...” – “tightly wound” is ambiguous in this context.

6) One thing missing from the discussion section is how the ssDNA secondary structure is melted, which is a prerequisite for homology sampling. Is it known if SSB is involved in the cell ? How does it work in the in vitro reaction, as the M13 DNA used would most likely have secondary structure. Could it be that local segments without secondary structure match first, followed by unraveling of the neighboring secondary structure ? Some mention of what is known in the literature would be helpful to complete and otherwise good discussion.

7) line 344: “splits right into the DNA strands” is unclear to me.

8) My only other comment pertains to the discussion of Rad52. I think it most likely that Rad52 works the same way as the authors do. However, the discussion of structural homology is distracting – the importance of the authors’ findings does not require extrapolation to Rad52. It is hard to know whether the similarities/differences represent convergent or divergent evolution, but the message one gets from the authors’ discussion is that Rad52 and RecT/Recbeta are evolutionarily related. I think it may be better to first point out all the structural similarities, including the 1st and 2nd DNA-binding grooves, then conclude that based on these similarities Rad52 is mechanistically likely very similar (and perhaps adding a statement about the uncertainty of convergent vs divergent evolution).

Caldwell et al., “Structure of a RecT/Red β family recombinase in complex with a novel duplex intermediate of DNA annealing”.

Point-by-point Response to Reviewers

Summary of changes: all of the changes are indicated in red. The changes include extensive new mutational data, as suggested by Reviewer #1. This is presented in two new figures (Fig. 5 & Fig. 6). Two new authors, Caroline Karbowski and Alyssa Wiegand, who performed the work for the mutational analysis, have been added. Though not mentioned by the reviewers, we have added a new supplementary Table 2 that compares the experimental mass of each LiRecT-DNA complex measured by native MS with the mass determined from the FW of the components. We have not made any changes to the structures themselves, which were deposited in the EMDB prior to our first submission. Finally, we note that a related work, a cryo-EM structure of an N-terminal fragment of λ -Red β bound to duplex DNA in a similar helical filament, has recently appeared in this journal (Newing et al., newly added reference 41). We have added a new paragraph at the end of the Discussion to briefly comment on this work. Here is a point-by-point response to all of the reviewers' comments:

Reviewer #1 (Remarks to the Author):

*“Single-stranded DNA (ssDNA) annealing (SSA) is involved in many DNA transactions to ensure genome integrity. It has been well documented that bacteriophage λ Red β , the SSA protein, binds weakly to ssDNA, not at all to preformed dsDNA, but has a higher affinity to a duplex intermediate of annealing formed when two complementary strands of ssDNA are added to the protein sequentially. However, no structural insights of such SSA proteins in complex with the intermediate state of annealing duplex DNA have been delineated. Here, the authors used a single particle cryo-EM approach to solve a 3.4 Å structure of a SSA protein LiRecT from prophage of *Listeria innocua* in complex with the annealing duplex DNA. The structural features of the LiRecT-dsDNA intermediate are revealed as follows: (1) LiRecT exhibits a left-handed helical filament with roughly 10 protomers per helical turn, (2) each monomer binds to 5 bp of dsDNA intermediates, (3) the structures of protomer-protomer and protomer-DNA interaction have been visualized and analyzed, and (4) the bound duplex DNA is highly extended and underwound compared to B-form DNA. In addition, the repeating 5 bases cause a large 9 Å spacing between bases. Finally, the cryo-EM results were verified in solution by native mass spectrometry. It is worth noting that, unlike Red β and Rad52 binding ssDNA as oligomeric rings, the author observed that the LiRecT-ssDNA complex exists in a left-handed helical filament structure.*

To my knowledge, this is the first structural documentation of the SSA protein with the intermediate state of annealing duplex DNA. Based on the structure information, the authors hypothesize that the N-terminal helix bundles of LiRecT clamp down on the duplex after the second strand is included to stabilize the complex and consolidate annealing. This provides a view of the SSA process at a mechanistic level. In addition, the structural information also revealed the structural similarity between LiRecT and eukaryotic Rad52, suggesting a possible SSA mechanism for Rad52. In summary, the novelty is well presented, and the manuscript is well written. However, there is no functional study to validate their structure observations. Addressing the following concerns would strengthen the evidence presented in the current manuscript:

1. It will be important to validate the structure findings via functional analyses. For example, the side chains of V98, Y100, and Q107 from the β 1- β 2 hairpin wedge into the base pairs at every 5th bp step to separate them (Fig. 2E). It will be interesting to make mutations in those residues to examine the functionality of strand annealing. In addition, given the structural homology with Rad52, it will be appreciated to conduct the functional study in either LiRecT or Rad52. For example, both proteins use the

conserved $\beta 1$ - $\beta 2$ hairpin to wedge into the DNA strands, and V98 from $\beta 1$ of LiRecT is precisely equivalent to R55 from $\beta 1$ of Rad52. Could we examine the effect of those mutant variants on SSA activity?"

RESPONSE: We have now added an extensive mutational analysis (starting on page 13 of the main text) that includes purification of 39 structure-based protein variants, and characterization of their ability to bind to ssDNA and form the complex with duplex intermediate. The mutations are targeted to four different regions of the protein: the site for binding the inner DNA strand, the outer DNA strand, the $\beta 1$ - $\beta 2$ hairpin wedge, and the inter-subunit interface. Many of the mutations at residues that contact the DNA (e.g. K206A) did not have strong effects. This is likely due to the fact that an extensive network of interactions is involved. Therefore we included several stronger mutations (e.g. K206E) as well as double-mutations (e.g. K206A/R210A) in our analysis. Overall, the effects of the mutations were not as strong as one might have expected, particularly those at the $\beta 1$ - $\beta 2$ hairpin wedge. This is likely due to the extensive network of interactions involved. We hope to address this more thoroughly and quantitatively with additional mutations in future studies. We did not include a mutational analysis of RAD52 that reviewer seemed to suggest, as we feel this is beyond the scope of the present paper, and many of the relevant RAD52 mutations have already been analyzed by others.

"2. In the cryo-EM structure of the LiRecT-ssDNA complex, the author could not see the ssDNA due to the low resolution of reconstruction. This significantly attenuates the importance of marking the structural differences between Rad52-ssDNA and LiRecT-ssDNA complexes. The authors should prove that a ssDNA indeed exists within the complex. Does LiRecT form a helical filament in the condition without DNA under cryo-EM?"

RESPONSE: First, we emphasize that there is density present in the expected site for binding ssDNA in the reconstruction for the 83-mer ssDNA complex, but the features just aren't clear enough for us to add it to the model (as they are for the backbone of several protein monomers). We have attempted to collect cryo-EM data from samples prepared for LiRecT without any added ssDNA. An example image from our 200 KeV Glacios instrument is now included as new Supplementary Figure 14 (and described in the main text at the bottom of page 11). We did not collect and process a full data set on the Krios, as the particles are much smaller than with the ssDNA complex, and barely visible – this observation is consistent with our native MS data of Fig. 4, which shows that LiRecT forms mostly monomer with a slight tendency to oligomerize. Overall, the particles seen by cryo-EM increase in degree of order in going from protein alone, to ssDNA complex, to complex with annealed duplex, precisely as indicated by the native MS data.

"3. Is any reason that the author used low pH 6 in cryo-EM sample preparation (page 19, line 423)? Does pH affect the LiRecT in complex with ssDNA and annealing duplex?"

RESPONSE: The buffer we chose for cryo-EM sample preparation (20 mM KH_2PO_4 , 10 mM MgCl_2 , pH 6.0) was adopted from the buffer that was used for early negative stain EM work showing rings and filaments for λ -Red β (Passy et al., reference 20). We have performed a new DNA-binding gel shift experiment comparing the cryo-EM buffer to PBS. The results are very similar. We have added the gels comparing DNA binding in both buffers (PBS and cryo-EM buffer) to revised Supplementary Figure 1 (the gel in PBS is also improved compared to the previous submission).

Reviewer #2 (Remarks to the Author):

"The paper by Caldwell et al presents substantial progress in the understanding of single strand

*annealing proteins (SSAPs). SSAPs were originally grouped together largely because they share some biochemical abilities to anneal DNA and also were not RAD51/RecA ATPases. The possibility that the SSAPs were related and form a RAD52 superfamily (named after the most important member) was proposed some time ago. This paper establishes the veracity of this important proposal and furthermore delivers remarkable mechanistic insight that will promote a great deal of progress. This is a remarkably significant paper that will be a milestone in the understanding of the poorly understood but crucial homologous recombination and DNA repair processes mediated by SSAPs. In particular, the details revealed by high resolution of the annealed DNA in the helical filament will motivate the debate surrounding RAD52 mechanism. Furthermore the diverse properties displayed amongst the vast array of prokaryotic SSAPs can now be evaluated with a new focus to consider key shared properties and diversifications. Since some of these prokaryotic SSAPs (*lambda* Red beta, *E.coli* RecT) are extremely useful for recombinant DNA and genome engineering, inventive applied progress is likely to emerge.*

The manuscript is superbly written at the highest standards. The data treatment is logical and thorough. I have only a few comments aimed at slight improvements.

Line 61, ref 22 is not the right reference and Datta et al, 2008 PNAS 105, 1626 would be better.”

RESPONSE: we have changed the reference to Datta et al. as suggested by the reviewer (page 4).

“Line 66, this sentence will be better as - "The structure of an 11-mer ring form of the DNA-binding domain of Rad52 was determined without DNA (28, 29) and with an oligonucleotide to form a substrate complex (30), but there is no ”.”

RESPONSE: we have made this change (middle paragraph, page 4).

*“Line 74, this sentence will be better as "The filaments are strikingly similar to those seen for *lambda*-Red beta at low resolution by electron microscopy and atomic force microscopy (20, 21), but our structure ”*

RESPONSE: we have made this change (last paragraph, page 4).

“Line 297, paragraph - the conclusions that the duplex intermediate formed by Red beta must lie on the outside of the helical filament and the DNA must be strongly unwound (not B-form) have been made before (ref 21).”

RESPONSE: We have added a new sentence to this paragraph of the Discussion (bottom of page 18), to describe the model proposed by Ander et al. (ref. 21). Note that this model included some features that are similar to our structure, and some that are different. While we do not provide a full comparison, we do refer readers to this model and do our best to briefly describe it and include the features that are similar.

“Line 321 - oligonucleotides as short as 35-mers are routinely used for Red beta recombineering in vivo.”

RESPONSE: we have added “oligonucleotides as short as 35-mers are routinely functional for λ -Red β annealing in vivo” to the discussion, and included an appropriate reference (last paragraph on page 19).

“The secondary structure labels on Figures 2A,B,C, 3A, SFig 4, SFig 10 are hard to read in places.”

RESPONSE: we have improved the secondary structure label positioning on all of these figures (and others) as much as possible and made the font larger.

“Figure 2 legend - denote the strand colours.”

RESPONSE: we have indicated the DNA strand colors in the legend.

Reviewer #3 (Remarks to the Author):

“The manuscript by Caldwell et al. reports the structure of a strand annealing reaction intermediate catalyzed by the RecT/Redbeta family of phage strand annealing proteins. There have been several structures of such proteins, including structures with a single ssDNA strand, but the structure of Caldwell et al. is the first instance when we see the annealed dsDNA. This structure represents a major advance in our understanding of single-strand annealing (SSA), a DNA repair process present in eukaryotes as well.

The RecT protein forms a left handed filament and binds to the 1st ssDNA as reported before (though some annealing proteins form rings). The complementary strand is held in place mostly through Watson-Crick hydrogen bonds, making annealing highly dependent on complementarity between the two strands. Globally, the DNA is stretched and under-wound compared to B DNA. Locally, however, the DNA is arranged in groups of 5 nucleotides; within each group the bases are well stacked, which is important for the Watson-Crick hydrogen bonds, while between groups there is a large spacing, suggesting complementarity sampling occurs in groups of 5 nts. These provide key mechanistic insights into the how proteins can catalyze SSA. Conceptually, they are similar to how the RecA and CRISPR Cascade proteins determine complementarity between two strands – a striking demonstration of how different proteins/processes can converge onto the same solution.

Structure-determination is solid. The ambiguity in what is happening at the ends of the filament is explained by flexibility there – a perfectly sensible explanation. The ambiguity in assigning the DNA sequence is also expected given that the filaments can stack end-to-end. None of these materially affect any of the conclusions.

Minor comments:

1) line 126: The term “common core” needs to be defined as the structural elements that align. It should also be indicated on Figs. 2A and 2B (or an adjacent figure). It may be easier to first define the RecT

secondary structure elements and then do the comparison; this way the common fold would be easier to explain. Also, an rmsd of 4.3 Å for 107 Cαphas (56% of RecT structure) would invariably contain spuriously overlapping residues (at least from a visual comparison of Figs 2A and B). The common core, as it is colored in Fig. S4, contains the beta sheet of strands 3/4/5, but in Fig 2B this sheet has an entirely different position, and while it has the same topology, structurally it is in a very different position; its residues should not be included spuriously in the rmsd measurement. Maybe the “common core” can be defined relative to alpha2/3 and beta1/beta2. A superposition of RecT and rad52 would also be very useful, though making it clear may be hard.”

RESPONSE: we have made it more clear in the text (page 7) that we have used the output from the pairwise superposition in the DALI server for defining the common core – Dali indicates, in the resulting sequence alignment that is output, the residues that are considered to align (based on an rmsd cutoff) – in this case the aligned residues include residues of the lower portions of the beta-345 sheet. We have added a new Supplementary Figure 4 that shows a stereo view of a superposition of LiRecT and Rad52 (panel A), as well as the regions on each individual structure that are considered to align (panels C and D). We have also added a new sentence to the main text (bottom of page7) to convey that the 345 sheet in LiRecT is much shorter than in Rad52, and that the upper portion of the beta-345 hairpin in Rad52 seems to be folded back over the sheet to form the betaAB insertion in LiRecT. Similarly the upper portion of the long beta-5 sheet seems to be replaced by the alpha-D helix in LiRecT. We hope that these further explanations, combined with the new figure showing the alignment and the core regions, and making it more explicit in the text that we are using the criteria output from DALI to determine the aligned regions, will make it more clear to the reader what we mean by the common core.

“2) Fig. 2B legend should state that Rad52 is oriented according to the superposition (if that’s indeed the case).”

RESPONSE: we have added a phrase to the Fig. 2 legend to indicate that the LiRecT and Rad52 monomers are shown in similar orientations. Since the protein structures are different we have chosen to show them in slightly different orientations, optimized for each to show as many features as possible, as opposed to the exact same orientation after alignment. Again, the new supplementary Fig. 4 shows a stereo view of the alignment, where the two are shown in exactly the same relative orientation, according to the superposition.

“3) I am not sure that discussion of RoseTTAFold adds anything of value, but I do not feel strongly about this (lines 145, 331, 339). It can just be stated that there is low-res density that could correspond to an N-terminal helix, but it is not interpretable (after all that is the data, and Rosetta is just prediction).”

RESPONSE: we have left the figure and description of the RosettaFold prediction in, since the reviewer seems OK either way, and the other reviewers did not comment on it. We feel that, though speculative, the prediction may provide some insights into N- and C-terminal regions of LiRecT that are presumably disordered in the structure. It is interesting that both proteins (LiRecT and λ-Redβ) are predicted to have an

additional downstream helix, right after the core fold, that binds within the DNA-binding groove, as if this helix might be bound for free monomers, before they assemble on the DNA (although this is speculative and we do not dwell on it).

“4) Is the DNA “under-wound” (line 171) or “completely unwound” (line 176) ? or do these statements refer to local and global DNA parameters ?”

RESPONSE: the DNA is completely un-wound – the two strands do not wrap around one another at all – the outer strand is always at a larger radius (on the outside) than the inner strand. We have changed all of the instances of “under-wound” to “un-wound” (all are highlighted in red).

“5) line 231: “tightly wound, with approximately 8 subunits per turn as opposed to ~10...” – “tightly wound” is ambiguous in this context.”

RESPONSE: we have omitted this sentence. It is a small difference and not particularly important. So that all of the reviewers can see the change, we have left it in, but highlighted it in red and put a slash through it.

“6) One thing missing from the discussion section is how the ssDNA secondary structure is melted, which is a prerequisite for homology sampling. Is it known if SSB is involved in the cell ? How does it work in the in vitro reaction, as the M13 DNA used would most likely have secondary structure. Could it be that local segments without secondary structure match first, followed by unraveling of the neighboring secondary structure ? Some mention of what is known in the literature would be helpful to complete and otherwise good discussion.”

RESPONSE: we have inserted a new paragraph into the Discussion, middle of page 20, to describe how we think DNA annealing occurs in the cell. Briefly, the exonuclease likely loads the recombinase directly onto the 3'-overhang as it is formed, before it folds into secondary structures. Then the recombinase-ssDNA complex likely interacts with the host SSB enzyme, to pair the first ssDNA with the second ssDNA at the target site of annealing as it is exposed on the lagging strand of the replication fork.

“7) line 344: “splits right into the DNA strands” is unclear to me.”

RESPONSE: we can see why this was confusing. We have changed this to “would overlap with the DNA if it were bound” (1st paragraph, page 21.). The helix predicted by RoseTTAFold occupies the same place in the DNA-binding groove that would be occupied by DNA.

“8) My only other comment pertains to the discussion of Rad52. I think it most likely that Rad52 works the same way as the authors do. However, the discussion of structural homology is distracting – the importance of the authors’ findings does not require extrapolation to Rad52. It is hard to know whether the similarities/differences represent convergent or divergent evolution, but the message one gets from the authors’ discussion is that Rad52 and RecT/Recbeta are evolutionarily related. I think it may be better to first point out all the structural similarities, including the 1st and 2nd DNA-binding grooves, then conclude

that based on these similarities Rad52 is mechanistically likely very similar (and perhaps adding a statement about the uncertainty of convergent vs divergent evolution)."

RESPONSE: we have changed the language to the paragraph where the relation to Rad52 is discussed (page 21), to "share a common core fold with Rad52" instead "share a related fold". We have similarly changed the language to "suggesting that there oligomers could be related", from "suggesting that their oligomers are related". We have been more careful throughout, including in the Abstract, to use terms such as "similar fold" instead of "related fold". We have not commented or expanded on the possibility/likelihood of convergent vs. divergent evolution.

REVIEWERS' COMMENTS

Reviewer #1 (Remarks to the Author):

In this revised manuscript, the authors have spent extensive efforts to conduct a mutational analysis. The results of this functional study correlate well with their structural observation. All the concerns raised by this referee have been fully addressed. The revised manuscript provides important insights into how proteins can catalyze SSA mechanistically. This referee strongly recommends this work to be published in Nature Communications.

Reviewer #2 (Remarks to the Author):

The original manuscript by Caldwell et al was excellent and important, and the revised manuscript is even more meritorious. Congratulations to the authors on their remarkable accomplishment, lucid manuscript and integrity.

Reviewer #3 (Remarks to the Author):

The authors have fully addressed my comments, which were minor to begin with. I was very supportive of the initial submission, and even more so of the revised submission. The revised ms certainly merits publication in Nature Communications.

Caldwell et al., “Structure of a RecT/Red β family recombinase in complex with an extended duplex intermediate of DNA annealing”.

Point-by-point Response to Reviewers

Reviewer #1 (Remarks to the Author):

“In this revised manuscript, the authors have spent extensive efforts to conduct a mutational analysis. The results of this functional study correlate well with their structural observation. All the concerns raised by this referee have been fully addressed. The revised manuscript provides important insights into how proteins can catalyze SSA mechanistically. This referee strongly recommends this work to be published in Nature Communications.”

RESPONSE: Thank you for your efforts in reviewing our manuscript, in particular for your suggestion for a mutational analysis, which we hope has strengthened the paper.

Reviewer #2 (Remarks to the Author):

“The original manuscript by Caldwell et al was excellent and important, and the revised manuscript is even more meritorious. Congratulations to the authors on their remarkable accomplishment, lucid manuscript and integrity.”

RESPONSE: Thank you for your efforts in reviewing our manuscript, in particular your suggestion of comparing to previous models of the Red β -DNA complex.

Reviewer #3 (Remarks to the Author):

“The authors have fully addressed my comments, which were minor to begin with. I was very supportive of the initial submission, and even more so of the revised submission. The revised ms certainly merits publication in Nature Communications.”

RESPONSE: Thank you for your efforts in reviewing our manuscript, in particular your suggestion to better delineate the portions of LiRecT and RAD52 that structurally align.